# Extrinsic Calibration of Thermal Camera and 3D LiDAR Sensor via Human Matching in Both Modalities during Sensor Setup Movement

**DOI:** 10.3390/s24020669

**Published:** 2024-01-20

**Authors:** Farhad Dalirani, Mahmoud R. El-Sakka

**Affiliations:** Computer Science Department, Western University, London, ON N6A 3K7, Canada; fdaliran@uwo.ca

**Keywords:** LiDAR, thermal camera, extrinsic calibration, sensor fusion

## Abstract

LiDAR sensors, pivotal in various fields like agriculture and robotics for tasks such as 3D object detection and map creation, are increasingly coupled with thermal cameras to harness heat information. This combination proves particularly effective in adverse conditions like darkness and rain. Ensuring seamless fusion between the sensors necessitates precise extrinsic calibration. Our innovative calibration method leverages human presence during sensor setup movements, eliminating the reliance on dedicated calibration targets. It optimizes extrinsic parameters by employing a novel evolutionary algorithm on a specifically designed loss function that measures human alignment across modalities. Our approach showcases a notable 4.43% improvement in the loss over extrinsic parameters obtained from target-based calibration in the FieldSAFE dataset. This advancement reduces costs related to target creation, saves time in diverse pose collection, mitigates repetitive calibration efforts amid sensor drift or setting changes, and broadens accessibility by obviating the need for specific targets. The adaptability of our method in various environments, like urban streets or expansive farm fields, stems from leveraging the ubiquitous presence of humans. Our method presents an efficient, cost-effective, and readily applicable means of extrinsic calibration, enhancing sensor fusion capabilities in the critical fields reliant on precise and robust data acquisition.

## 1. Introduction

The challenges encountered in the realm of computer vision often present a high degree of complexity. To address these complexities effectively, it is common to employ a range of sensors that work collaboratively to augment the information gathered from the scene and the objects within it. The integration of diverse sensors frequently leads to solutions that not only enhance accuracy but also bolster robustness [1]. 3D LiDAR (Light Detection and Ranging) sensors and thermal cameras, valued for their accurate point clouds and heat information receptivity, are gaining attention for use in data fusion. Extrinsically calibrating these sensors, each with its own coordinate system, is essential for their accurate data integration.

3D LiDAR sensors have emerged as one of the most popular sensors in fields such as agriculture, autonomous vehicles, and robotics. Some of their applications include odometry and SLAM (Simultaneous Localization and Mapping) [2] in robotics, semantic scene understanding [3], and 3D object detection [4] in self-driving cars, forest attribute estimation [5], and precision farming [6].

A LiDAR sensor produces a 3D point cloud where each point is precisely defined by its x, y, and z LIDAR coordinates. Furthermore, this point cloud includes data regarding the strength of the reflected laser pulse at each point. Consequently, a LiDAR sensor does not offer supplementary information for individual points, such as color. However, when we integrate LiDAR data with additional data from other sensors, it becomes feasible to improve performance across a range of tasks. For instance, in the study by Xu et al. [7], LiDAR data was combined with data from an RGB camera to enhance 3D object detection.

Thermal cameras have gained attention as alternative sensors to fuse with LiDAR data due to their ability to create high-quality images based on temperature differences in objects and their surroundings, even in adverse conditions like darkness, snow, dust, smoke, fog, and rain [8]. Because thermal cameras can capture spectra that other sensors like visual light cameras cannot, they have numerous applications in agriculture, security, healthcare, the food industry, aerospace, and the defense industry, among others [9,10].

Combining data from 3D LiDAR sensors and thermal cameras can yield the benefits of both sensors simultaneously. By leveraging both 3D spatial information and heat signatures, a more comprehensive and accurate representation of the environment is achieved. This integration enhances overall situational awareness, robustness, and accuracy across many tasks, especially when compared with the use of either technology in isolation. For example, in any application involving the heat data of a scene and its objects, it can be augmented with LiDAR data to obtain the 3D location of various elements within the scene. For instance, when measuring the attributes of fruits on a tree or detecting pedestrians in the streets, leveraging the 3D location can provide accurate positioning information to allow the robotic arm to harvest the fruit or enable the control component in an autonomous vehicle pipeline to take necessary actions to avoid colliding with pedestrians. The following are some of the existing applications of combining these two sensors for various purposes. Kragh et al. [11] instrumented a tractor with multi-modal sensors, including LiDAR and a thermal camera, to detect static and moving obstacles, including humans, to increase safety during operations in the field. Choi et al. [12] developed a multi-modal dataset including LiDAR and thermal camera data for studying various tasks, including drivable region detection, object detection, localization, and more, in the context of assisted and autonomous driving, both during the day and at night. Shin et al. [13] used LiDAR and thermal cameras to investigate depth estimation in challenging lighting and weather conditions for autonomous vehicles. In their research, Yin et al. [14] built a ground robot instrumented with various sensors, including a thermal camera and LiDAR. They argued that visual SLAM with an RGB camera is ineffective in low visibility situations such as darkness and smoke, and using a thermal camera can address some of these challenges. Tsoulias et al. [15] used a thermal camera and LiDAR to create a 3D thermal point cloud to detect disorders caused by solar radiation on fruit surfaces. Yue et al. [16] incorporated a thermal camera alongside LiDAR to enhance the robots’ ability to create a map of the environment, both during the day and at night.

A thermal camera and LiDAR have their own coordinate systems. To use data from both modalities, these two sensors should be extrinsically calibrated. Here, extrinsic calibration is the task of finding the rotation matrix R and translation vector t to express the coordinate of a point in the LiDAR’s coordinate system in the camera’s coordinate system. R is an orthogonal 3×3 matrix that describes rotation in 3D space, and t is a 3D vector that represents a shift in 3D space. After obtaining the extrinsic parameters, the point $p^{C}$ in the thermal camera system corresponding to the LiDAR point $p^{L}$ in the LiDAR coordinate system can be obtained according to $p^{C}=Rp^{L}+t$.

In the extrinsic calibration of visible light cameras and LiDAR, various types of targets, including checkerboard targets [17], are typically employed. Nonetheless, these targets are not visible to a thermal camera. To adapt them for the extrinsic calibration of a thermal camera and LiDAR, these targets can be modified by crafting them from various heat-conductive materials and then either pre-cooling or heating them before use [18], or by incorporating heat-generating electrical elements such as light bulbs [15]. Using these adopted targets comes with some drawbacks. Creating them is both challenging and expensive. Using them in situations where the sensor setup frequently changes or sensor drift occurs can be cumbersome. Additionally, over time, heating leaks can occur from the heat-generating elements, or their temperature can become similar to the surrounding environment, rendering them ineffective for use, and getting them operational again can take some time.

The mentioned difficulties encountered while working with calibration targets motivated our proposed method. We propose a novel method for the extrinsic calibration of a thermal camera and a LiDAR without using a dedicated calibration target based on matching segmented people in both modalities during the movement of the sensor setup in environments such as farm fields or streets that contain humans. The extrinsic parameters are obtained by optimizing a designed loss function that measures the alignment of human masks in both modalities. This is achieved using a novel optimization algorithm based on evolutionary algorithms. We present two versions of our algorithm. The first version disregards input noise, while the second version seeks to mitigate the effects of noisy inputs. This innovative approach minimizes the expenses associated with the creation of calibration targets for thermal cameras and eliminates the often labor-intensive and time-consuming process of collecting diverse poses for calibration targets, particularly in the context of autonomous vehicles where positioning a large target at various angles and heights can be challenging. It also addresses the issue of the repetitive calibration efforts required when sensor drift or setting changes occur, making the process more efficient. Additionally, it enhances the accessibility of 3D LiDAR and thermal camera fusion by eliminating the necessity for specific targets.

The remainder of the paper is structured as follows: In Section 2, we provide an overview and examination of prior research. Section 3 outlines the cross-calibration algorithm. Section 4 showcases our experiments and their outcomes on the FieldSAFE [11] and ${\mathrm{MS}}^{2}$ [13] datasets. Lastly, Section 5 serves as the conclusion of our paper and outlines potential avenues for future research.

## 2. Related Work

Some studies have explored the calibration of thermal cameras and LiDAR systems using various target-based approaches. These methods typically involve utilizing the known specifications of the calibration targets and minimizing a cost function to establish the extrinsic parameters that align these specifications across both sensor modalities. Krishnan et al. [18] used a checkerboard target made of laser-cut black and white melamine with different heat conductivity. They placed it in front of the sun for approximately one hour to enable the detection of checkerboard corners by a thermal camera. A user manually selected the four outer corners of the target inside the thermal image, and to detect the calibration target within the point cloud, they used a region-growing algorithm. They determined the rotation matrix and translation vector by attempting to minimize the distance between the points on the edges of the target in the LiDAR point cloud and their nearest points on the edges of the target in the thermal image. Their algorithm requires a good initial rotation, translation, and several poses. Krishnan et al. [19] developed a cross-calibration method that involved the creation of a target by cutting a circular hole in white cardboard with a precisely known radius. They utilized a damp black cloth as the background, which improved the circle’s visibility in the thermal camera. The process started by manually selecting a pixel in the circle for a region-growing algorithm to segment it in the image. Likewise, the user picked a point on the cardboard to locate the target in the point cloud. They captured multiple poses for cross-calibration. In each pair, they projected the circle’s edges from the point cloud onto the thermal image. Finally, they solved an optimization problem of aligning the thermal camera’s circle edges with the projected edges, ensuring precise calibration. Borrmann et al. [20] devised a calibration target visible in thermal cameras by creating a dot pattern on a board using light bulbs. In the calibration process, they collected multiple pairs of images and their corresponding point clouds. For each of these pairs, they precisely determined the locations of the light bulbs in in both modalities. To establish the positions of the light bulbs within the LiDAR coordinate system, they located the calibration target within the point cloud data. Leveraging the well-defined geometry of their calibration target, they computed the positions of the light bulbs in the LiDAR coordinate system. Subsequently, for each image-point cloud pair, they mapped the positions of the light bulbs from the point cloud to the thermal image. Finally, to determine the extrinsic parameters, they solved an optimization problem aimed at minimizing the disparity between the light bulb positions in the thermal image and their projected positions in the point cloud. In the proposed method of Dalirani et al. [21], an active checkerboard target with embedded resistors for generating heat was used, and extrinsic parameters between both the thermal and LiDAR sensors were obtained from the correspondence of lines and plane equations of the calibration target in the image and point cloud pair. Zhang et al. [22] created four equally spaced circles on an electric blanket. They identified these circles in both modalities and optimized the extrinsic parameters by minimizing the 2D re-projection error.

In many studies, when using a thermal camera and LiDAR data, instead of directly performing extrinsic calibration between the thermal camera and LiDAR, each of them is extrinsically calibrated with another sensor, such as an RGB camera, for example. Then, the two sets of obtained extrinsic calibration parameters are used to determine R and t between the thermal camera and LiDAR. Azam et al. [23] employed a thermal camera capable of providing both visual and thermal images, along with extrinsic parameters linking these two types of images. They applied an established RGB camera-LiDAR calibration technique to achieve extrinsic calibration between the visual camera and LiDAR. Subsequently, they utilized this knowledge, in conjunction with extrinsic calibration parameters connecting the visual and thermal cameras, to derive the transformation between the thermal camera and the LiDAR. Similarly, Zhang et al. [24] divided the calibration process for the thermal camera and LiDAR into two sequential steps. In the FieldSAFE dataset [11], a similar method [25] was employed to determine the rotation and translation between sensors. They calculated the extrinsic parameters between the LiDAR and the stereo vision system using the iterative closest point algorithm [26]. To calibrate the stereo vision system and the thermal camera, they constructed a checkerboard with both copper and non-copper materials and attached 60 resistors to generate heat. Subsequently, through post-processing, they were able to employ a regular cross-calibration tool for two visual light cameras to extrinsically calibrate the RGB and thermal cameras. Finally, by comparing the two solutions, the parameters between the thermal and LiDAR sensors could be obtained. In the ${\mathrm{MS}}^{2}$ dataset [13,27], for their instrumented car, they established extrinsic calibration parameters between all sensors, including the thermal cameras and LiDAR, in conjunction with the NIR camera. The rotation and translation between other sensors can be obtained by using these extrinsic parameters with the NIR camera. To calibrate the NIR and thermal cameras, they used a 2×2 AprilTag board with metallic tape attached to it.

In another approach, targetless extrinsic calibration methods do not use a target but instead employ feature alignment in both modalities. Fu et al. [28] introduced a targetless extrinsic calibration method that calibrates a stereo visual camera system, a thermal camera, and a LiDAR sensor. In their method, first, the transformation between LiDAR and the stereo system is estimated. Then, the thermal camera is calibrated with the left camera in the stereo system by simultaneously using data from LiDAR and the left stereo camera. By establishing transformations between the thermal camera and the stereo system, as well as between LiDAR and the stereo system, the transformation between LiDAR and the thermal camera can be calculated. Their method optimizes extrinsic parameters by maximizing the alignment of edges in the three modalities. To derive edges from the LiDAR point cloud, they employed the horizontal depth difference and utilized the Canny edge detector [29] to detect edges in the thermal camera and the left stereo camera. Their method requires sufficient edge features in the modalities and a rough initial guess for optimization. Mharolkar et al. [30] proposed a targetless cross-calibration method for visual and thermal cameras with LiDAR sensors by utilizing a deep neural network. Instead of employing hand-crafted features, they utilized multi-level features from their network and used these extracted feature maps to regress extrinsic parameters. To train the network for calibrating the visual camera and LiDAR on the KITTI360 dataset [31], they utilized 44,595 image-point cloud pairs. For training the network for calibrating the thermal camera, they employed pre-trained weights for the visual camera and LiDAR and trained the model on their thermal camera and LiDAR dataset, consisting of 8075 thermal images and LiDAR pairs. Additionally, for a new set of sensors, the network should be re-trained.

Our proposed method does not require a target and optimizes extrinsic parameters during the movement of the sensor setup in an environment with human presence by aligning segmented people in both modalities. Importantly, it does not rely on the presence of rich edge features, making it applicable even in environments like farm fields, which often lack distinct edges. Moreover, it does not demand a precise initial solution, enhancing its versatility and ease of use. To the best of our knowledge, in the literature on the extrinsic calibration of thermal cameras and 3D LiDAR sensors, our proposed method represents a novel approach distinct from any existing methodologies. To date, no other method for these sensors has demonstrated the same innovative techniques employed in our study.

## 3. Methodology

In this paper, we propose an extrinsic calibration method for determining rotation matrix R and translation vector t between a thermal camera and a 3D LiDAR sensor without the need for a target. Our method relies on matching segmented humans in both modalities during the movement of the sensor setup. In the following, we will explain the steps of the proposed method, including data collection, formulating the problem, designing a cost function, and the method for optimizing the extrinsic parameters by minimizing the cost function.

### 3.1. Data Collection

While the thermal camera and LiDAR sensor setup is in motion on a moving vehicle, such as a tractor, robot, or car, in various environments like streets and farm fields, the dataset D is created by capturing several frames at different time points, denoted as $t_{1}$, $t_{2}$, …, $t_{N_{pose}}$, for both modalities. $N_{pose}$ denotes the number of captured frames. At each time $t_{i}$, both the LiDAR and thermal camera capture the scene simultaneously, producing the captured image and point cloud, which we denote as $I_{t_{i}}$ and $P_{t_{i}}$, respectively. Given that our method relies on matching humans in both modalities, it is essential that each image and point cloud pair in the dataset contains human subjects, and the number of humans should be equal, which may vary from one or more individuals. As the number of humans increases, the likelihood of overlapping also rises, introducing more errors in segmenting humans in both data modalities. Therefore, only frames containing between one and a small number, denoted as $H_{max}$, of humans are retained.

In the beginning, the dataset *D* is empty. During the movement of the sensor setup in the environment at the moment $t_{i}$, a thermal image and a point cloud are captured. Then, an off-the-shelf person segmentation model and a human detector are applied to the captured image and point cloud, respectively. If the number of humans found in both modalities is equal and is greater than zero, the image and point cloud pair are kept; otherwise, it is discarded. In the provided pair, $I_{t_{i}}^{h}$ is generated by assigning a value of one to pixels within the human masks and zero to pixels outside the masks in the thermal image. Similarly, $P_{t_{i}}^{h}$ is produced by retaining the points in the point cloud that correspond to humans and removing all other points. Subsequently, the $I_{t_{i}}^{h}$ and $P_{t_{i}}^{h}$ pair is included in the dataset *D*. This process continues until the dataset *D* reaches a specific size, denoted as $N_{pose}$. Two examples from the FieldSAFE [11] and ${\mathrm{MS}}^{2}$ [13] datasets are shown in Figure 1.

In collected data pairs, one important consideration is that humans should be positioned at various locations and sizes within the thermal image. Otherwise, the obtained extrinsic parameters will exhibit bias toward specific areas, causing them to deviate from the actual parameters. Furthermore, since the positions of humans in both thermal images and point clouds do not change significantly in consecutive frames, when a thermal image and point cloud are added to the dataset at time $t_{i}$, the next three frames will not be considered for inclusion in the dataset.

### 3.2. Cost Function

To optimize the extrinsic parameters R and t between a thermal camera and a 3D LiDAR sensor, based on human matching in both modalities, a cost function is required to measure the alignment of humans in both modalities for all thermal image and point cloud pairs ($I_{t_{i}}^{h}$, $P_{t_{i}}^{h}$) in the dataset D, with respect to a set of extrinsic parameters.

When provided with a candidate rotation matrix R and a translation vector t for image and point cloud pairs ($I_{t_{i}}^{h}$, $P_{t_{i}}^{h}$), the loss is calculated according to Equation (Equation 1).
(1)$$Loss\left(I_{t_{i}}^{h},P_{t_{i}}^{h};R,t\right)=\frac{1}{|P_{t_{i}}^{h}|}\sum_{p^{L}\in P_{t_{i}}^{h}}\psi \left(K\left(Rp^{L}+t\right);I_{t_{i}}^{h}\right)$$

In Equation (Equation 1), $p^{L}$ iterates points in the point cloud $P_{t_{i}}^{h}$, K is the 3×3 intrinsic camera matrix, and |·| denotes the number of points in a point cloud. In this equation, $Rp^{L}+t$ maps the point $p^{L}$ from the LiDAR coordinate system to camera coordinate ($p^{C}$), and multiplying it by K maps the point to camera image coordinate ($p^{I}$). $p^{I}$ is inhomogeneous representation and should be converted to inhomogeneous. $\psi (p^{I};I_{t_{i}}^{h})$ is a function that outputs a penalty score based on distance of the projected point $p^{I}$ from LiDAR coordinate system to image coordinate to the nearest human pixel in $I_{t_{i}}^{h}$. The function ψ is defined according to Equation (Equation 2).
(2)$$\psi \left(p^{I};I_{t_{i}}^{h}\right)=\left\{\begin{array}{cc} {∥p^{I}-p_{near}∥}_{1} & \mathrm{if}\;p^{C}\;\mathrm{is}\;\mathrm{in}\;\mathrm{front}\;\mathrm{of}\;\mathrm{the}\;\mathrm{camera}\;\mathrm{image}\\ c_{1}\times max\left(h\left(I_{t_{i}}^{h}\right),w\left(I_{t_{i}}^{h}\right)\right) & \mathrm{if}\;p^{C}\;\mathrm{is}\;\mathrm{behind}\;\mathrm{the}\;\mathrm{camera}\;\mathrm{image} \end{array}\right.$$

In Equation (Equation 2), ${∥\cdot ∥}_{1}$ represents the Manhattan distance, and h(·) and w(·) provide the height and width of $I_{t_{i}}^{h}$. Additionally, $p_{near}$ represents the nearest human pixel in $I_{t_{i}}^{h}$ to $p^{I}$. ψ is a piecewise function. If a projected point from the LiDAR coordinate system to the camera coordinate system is in front of the camera, the function calculates the distance of the point projection in the thermal image coordinate system to the nearest human pixel. If the projected point from the LiDAR coordinate system to the camera coordinate system is behind the camera, it indicates that the projection is highly invalid. In such cases, we impose a significant penalty by assigning a large value. We determined this penalty to be the maximum value between the image height and width, multiplied by the constant $c_{1}$. Selecting a low value, such as one for $c_{1}$, means that we do not differentiate enough between a mapping that projects a LiDAR point in front of the camera, outside the image, and not too far from the edges of the image, and a mapping that projects the LiDAR point to the back of the camera. A larger value of $c_{1}$, such as five, makes cases like this more distinguishable. In Figure 2, the loss for two sets of extrinsic parameters for one pair of thermal images and point clouds from the FieldSAFE dataset [11] is shown. The loss for Figure 2a is 1.35, which is much smaller than the 58.38 loss for Figure 2b. In the case of Figure 2b, greater deviations in the extrinsic parameters from the true values caused LiDAR-projected points to be further from humans in the image, resulting in a larger loss.

The total loss for a candidate R and t on dataset D is the average of losses on all image and point cloud pairs in the dataset, as defined in Equation (Equation 3).
(3)$$Loss\left(D;R,t\right)=\frac{1}{\left|D\right|}\sum_{(p_{I}^{L},I_{t_{i}}^{h})\in D}Loss\left(I_{t_{i}}^{h},P_{t_{i}}^{h};R,t\right)$$

### 3.3. Optimization Method

In the proposed method, the estimate of the extrinsic parameters, R and t, that describes the relationship between a thermal camera and a LiDAR sensor, involves the minimization of Equation (Equation 3). To achieve this, we introduced an optimization approach rooted in evolutionary algorithms for the purpose of parameter calculation between these two sensors. Since errors, such as false positives, false negatives, under-segmentation, and over-segmentation, can occur in the detection and segmentation of humans in both modalities, the proposed algorithm incorporates a mechanism to reduce the effect of outliers. First, we will explain the algorithm that does not consider outlier rejection, Algorithm 1. Afterward, we will provide a comprehensive explanation of the Algorithm 2.

We decided to create the optimization algorithm based on evolutionary algorithms for the following reasons. First, in the case of non-differentiable or noisy objective functions, evolutionary optimization can obtain good solutions. Second, evolutionary optimization is much less likely to be affected by local minima, and it eliminates the need for an initial solution in our calibration method. Third, evolutionary algorithms often exhibit greater robustness in the face of noisy and uncertain observations.

Algorithm 1 presents the proposed algorithm, omitting any outlier rejection. The algorithm creates a population of random individuals and gradually evolves the population in each generation to optimize R and t. Each individual of the population is an instance of Individual structure. As demonstrated in lines 1–6 Algorithm 1, the Individual structure consists of four fields. The first field, denoted as *t*, represents the translation vector from a LiDAR sensor to a thermal camera. The second field, labeled as *r*, corresponds to Rodrigues’ rotation vector from the LiDAR to the thermal camera. Instead of directly optimizing the rotation matrix R with its 9 elements and managing its orthogonality, we optimize rotation vector r with only 3 parameters and subsequently convert it to rotation matrix R using OpenCV’s Rodrigues function [32]. The third field comprises the resulting loss on the dataset based on the individual’s *r* and *t*, which is calculated according to Equation (Equation 3). The fourth field for an individual represents the probability of selection for crossover and mutation, a concept we will elaborate on further.
**Algorithm 1** Proposed algorithm without outlier handling**Require:** *D*, $N_{pop}$, $iter_{max}$, $interval_{rot}$, $interval_{tran}$, $pct_{elite}$, $pct_{crossover}$ 1:**Struct** Individual { 2:     vector3D *t*; 3:     vector3D *r*; 4:     float loss; 5:     float prob; 6:} 7:population = initialPopulation(size=$c_{2}\times N_{pop}$, $interval_{rot}$, $interval_{tran}$) 8:**for** $iter_{i}=1$ to $iter_{max}$ **do** 9:   nextPopulation = {}10:   **if** $iter_{i}>1$ **then**11:     population = top $N_{pop}$ lowest loss individuals in population12:   **end if**13:   **for** individual in population **do**14:     individual.loss = Loss(*D*; Rodrigues(individual.r), individual.t) (Equation (Equation 3))15:   **end for**16:   **for** individual in population **do**17:     individual.prob = selectionProbability(population, individual)18:   **end for**19:   Add the top $\left(pct_{elite}\times |population|\right)$ lowest loss individuals to nextPopulation.20:   Randomly select $\left(pct_{crossover}\times |population|\right)$ pairs with replacement from population based on the probability of each individual.21:   Apply the ‘crossOver()’ operation to each selected pair and add the resulting new individuals to the nextPopulation.22:   Randomly select (|population|−|nextPopulation|) individuals with replacement from the population based on the probability of each individual.23:   Apply the ‘mutation()’ operation to each selected individual and add the resulting new individuals to the nextPopulation.24:   population←nextPopulation25:**end for**26:**return** R and t of individual in population with smallest individual.loss

This algorithm operates on a dataset denoted as *D*, which has been generated in accordance with Section 3.1. It takes parameters like $N_{pop}$, signifying the number of individuals in the population, and $interval_{rot}$ and $interval_{tran}$, representing the rotation and translation intervals used for generating random initial individuals in the population. Furthermore, we have $pct_{elite}$, a parameter that determines the percentage of the best-performing individuals with the lowest loss to be retained in the next generation. Additionally, $pct_{crossover}$ is another parameter that specifies the percentage of the population selected for crossover.

In line 7, the initial population is generated using the ‘initialPopulation’ function. To enhance diversity, the size of the population that it generates is set to be $c_{2}$ times larger than $N_{pop}$. However, after the first iteration, the population size is reduced to $N_{pop}$, as shown in lines 10–12. If the number of individuals in the population is low, setting $c_{2}$ to a value like five can increase diversity. However, when the population is large, it can be set to one to prevent unnecessary computation. To create a random individual within the population, ‘initialPopulation’ initializes an instance of the Individual structure. The function randomly samples all three elements of vectors *t* and *r* from the intervals $interval_{tran}$ and $interval_{r}$, respectively. In all our experiments, we assumed no prior information about the LiDAR and thermal camera position and orientation relative to each other. We selected a wide interval of [−3.5, 3.5] radians and [−1, 1] meters; however, a user can choose smaller intervals if they wish to incorporate prior knowledge about the positions and orientations of sensors. Next, the produced individual becomes part of the population under the condition that, for a pair of $I_{t_{i}}^{h}$ and $P_{t_{i}}^{h}$ in dataset *D*, a minimum of 50% of the points in $P_{t_{i}}^{h}$ project within the thermal image. This projection is achieved through the utilization of a randomly generated rotation vector *r* and translation vector *t* associated with the individual. In case this criterion is not met, the individual is discarded, and a new one is generated in its place.

Between lines 8 and 25, the next generation is formed through a process that combines elitism, crossover, and mutation techniques. In lines 13–15, the loss on dataset *D* for each individual is computed as per Equation (Equation 3). In lines 16–18, individual.prob is calculated for each individual in the population using the ‘selectionProbability’ function as defined in Equations (Equation 4) and (Equation 5). The first one computes a fitness score based on individual loss relative to the population, and the second one calculates the selection probability for an individual, taking their fitness score and the sum of fitness scores for the entire population into account.
(4)$$individual.score=1-\frac{individual.loss}{\sum_{ind\in population}ind.loss}$$
(5)$$individual.prob=\frac{individual.score}{\sum_{ind\in population}ind.score}$$

In line 19, the top $pct_{elite}$ percent of individuals with the lowest loss in the population are directly copied to the next generation. This elitism ensures that the best solutions found so far are not lost and continue to contribute to the population’s overall quality over the next generations.

Between lines 20–23, individuals for crossover and mutation are selected, and the functions ‘crossOver’ and ‘mutation’ are applied. ‘crossOver’ creates a new individual from a pair of individuals according to Equations (Equation 6) and (Equation 7). In these two equations, individualOne and individualTwo are two members of the population, and individualOne has a lower loss than the other one. Also, α is a random number between 0.5 and 1. The function ‘mutation’ affects an individual by applying noise to its rotation and translation vectors, creating a new individual. The ‘mutation’ operation adds random uniform noise within the range of $\left[-\sigma_{rot},\sigma_{rot}\right]$ to each element of the rotation vector and independently adds noise within the range of $\left[-\sigma_{trans},\sigma_{trans}\right]$ to each element of the translation vector.
(6)newIndividual.r=α·individualOne.r+(1−α)·individualTwo.r
(7)newIndividual.t=α·individualOne.t+(1−α)·individualTwo.t

Algorithm 2 contains the complete proposed algorithm, which attempts to mitigate the effects of outlier data pairs. The general idea of this algorithm is to handle outliers in a dataset (*D*) by iteratively fitting a model to a small subset of the data, identifying and removing outliers based on a loss threshold, and then re-fitting the model to the inliers. The algorithm is designed to robustly estimate rotation (R) and translation (t) parameters for a given dataset.

Algorithm 2 requires all the inputs of Algorithm 1, with the addition of some extra inputs. $min_{sample}$ represents the size of a random subset of *D* that is chosen to find extrinsic parameters. $iter_{outlier}$ denotes the number of fitting attempts to detect outliers. $threshold_{sample}$ determines whether a sample should be considered an outlier or not. If the calculated loss for a sample pair, as per Equation (Equation 1), is greater than $threshold_{sample}$, it is considered an outlier. A solution of a fitting attempt on the selected subset of *D* is deemed correct if the ratio of samples with a loss smaller than or equal to the value of $threshold_{sample}$ is greater than or equal to $ratio_{solution}$. Furthermore, I(·) represents the indicator function. It outputs the value of one when the condition is met and zero otherwise.
**Algorithm 2** Proposed algorithm with outlier handling**Require:** *D*, $N_{pop}$, $iter_{max}$, $interval_{rot}$, $interval_{tran}$, $pct_{elite}$, $pct_{crossover}$, $min_{sample}$, $iter_{outlier}$, $ratio_{solution}$, $threshold_{sample}$1:Create an array, isInlier, with a size of |D| and initialize each element with True2:**for** $iter_{i}=1$ to $iter_{outlier}$ **do**3:   Create $D_{train}$ by randomly sampling $min_{sample}$ data pairs from *D*4:   Create $D_{val}$ using the remaining data pairs from *D*5:   Obtain R and t by using Algorithm 16:   listLosses = loss of R and t for each data pairs in $D_{val}$ using Equation (Equation 1)7:   $ratio_{inliers}=\frac{\sum_{a\in listLosses}I\left(a<=threshold_{sample}\right)}{\left|ratio_{inliers}\right|}$8:   **if** $ratio_{inliers}>=ratio_{solution}$ **then**9:     **for** $pair_{i}$ in $D_{val}$ **do**10:        **if** $listLosses[pair_{i}]$>$threshold_{sample}$ **then**11:          $isInlier[pair_{i}]\leftarrow$ False12:        **end if**13:     **end for**14:   **end if**15:**end for**16:Create $D_{inlier}$ by selecting elements in *D* where the corresponding element in $isInlier[pair_{i}]$ is True17:Obtain R and t by applying Algorithm 1 to $D_{inlier}$18:**return** R and t

The proposed algorithms aim to determine a rigid body transform between the coordinate systems of a thermal camera and a LiDAR sensor by estimating the rotation matrix R and translation vector t. It is essential for both sensors to operate with the same scale for accurate results. If the two sensors are not on the same scale, and assuming the factory configurations of sensors are available (which is almost always the case for these two types of sensors), this information can be used to preprocess the data and convert them to the same scale. In Equation (Equation 1), $K(Rp^{L}+t)$ is utilized to map a LiDAR point in the image coordinate system in a homogeneous format. Subsequently, the homogeneous point is converted to an inhomogeneous coordinate in the thermal image. When using data with different scales, as the cost function minimizes the distance in the thermal image, it can yield a solution that effectively maps LiDAR points to their corresponding thermal image pixels, even when dealing with data of varying scales. However, the obtained translation vector may not accurately represent the real distance between the sensors, as it will be scaled by the difference in scale between the two sensors.

To efficiently calculate the function in Equation (Equation 1), for each $I_{ti}^{h}$ in a collected dataset, an array with a height of *h* and a width of *w* can be created, where each element represents the distance from that pixel to the nearest pixel belonging to a human. For a dataset of size |D|, the computational complexity of this operation is O(|D|.w.h). In Equation (Equation 2), for a given $I_{t_{i}}^{h}$ and $P_{t_{i}}^{h}$ pair, for the number of points in the point cloud $\left(|P_{t_{i}}^{h}|\right)$, several fixed matrix multiplications and summations take place. Therefore, for one pair, the computational complexity will be $O(|P_{t_{i}}^{h}|)$. According to Equation (Equation 3), its computation complexity is $O(|D|.|P_{t_{i}}^{h}|)$. Therefore, since Algorithm 1 performs $iter_{max}$ iterations, and each iteration calculates the loss of individuals on a scale of $N_{pop}$, the total computational complexity will be $O(|D|\cdot w\cdot h)+O(|D|\cdot |P_{max}^{h}\left|\cdot N_{pop}\cdot iter_{max}\right)$, where $|P_{max}^{h}|$ is the number of points in the point cloud with the most points. The computational complexity of Algorithm 2 remains the same, with the additional step of calculating extrinsic parameters using a subsampled dataset of size $min_{sample}$ for $iter_{outlier}$ times.

## 4. Experiments

To evaluate our method, we used the FieldSAFE dataset [11] and the first sequence of the ${\mathrm{MS}}^{2}$ dataset [13]. The selection of this sequence was random, as it is assumed to be representative of the dataset, given that the sensor setup is identical across all sequences. The FieldSAFE dataset [11] contains data from a tractor equipped with various sensors, including a thermal camera and a LiDAR sensor, captured during a grass-mowing scenario in Denmark. The ${\mathrm{MS}}^{2}$ dataset comprises data collected by an instrumented car with different sensor types, such as a thermal camera and LiDAR sensor, in various environments, including city, residential, road, campus, and suburban areas. The thermal camera in the FieldSAFE dataset is a FLIR A65 with a maximum frame rate of 30 frames per second (FPS) and a resolution of 640 × 512 pixels. It obtained LiDAR data from the Velodyne HDL-32E, which is a 32-beam LiDAR sensor with a 10 FPS data rate and 2 cm accuracy. The thermal camera in the ${\mathrm{MS}}^{2}$ dataset is the same as in FieldSAFE, and the LiDAR is a Velodyne VLP-16, which has sixteen LiDAR beams, a maximum frame rate of 20 FPS, and 3 cm accuracy. In the ${\mathrm{MS}}^{2}$ dataset, the provided thermal images have a resolution of 640 by 256 pixels. Moreover, in both datasets, the positions and orientations of the sensors with respect to each other are highly different. Our proposed algorithm produces accurate results on both setups, including sparse 16-beam and dense 32-beam LiDARs, demonstrating its effectiveness. Also, in both datasets, the intrinsic camera matrices (K) of thermal cameras are available.

We created two datasets from FieldSAFE and ${\mathrm{MS}}^{2}$ following the guidelines in Section 3.1. Additionally, we generated two other datasets for evaluation purposes by manually selecting and annotating the data. For human segmentation in thermal camera images, we utilized Faster R-CNN [33] trained on a FLIR thermal dataset [34] and subsequently fed the bounding boxes into the Segment Anything Model (SAM) [35]. To extract humans from the LiDAR point cloud, we employed MMDetection3D [36]. The dataset created from FieldSAFE consists of 63 training examples and 20 test samples, while the dataset extracted from ${\mathrm{MS}}^{2}$ comprises 55 training examples and 19 test samples. For simplicity, we denote them as $D_{FS}^{train}$, $D_{FS}^{test}$, $D_{MS}^{train}$, and $D_{MS}^{test}$. Since there are often only one to three persons in the sequences used from both the FieldSAFE and ${\mathrm{MS}}^{2}$ datasets, we selected $H_{\max}$ to be equal to three. In $D_{FS}^{train}$, the mean spatial location of all humans in thermal images is (305.82, 103.49), with standard deviations of 155.9 and 43.3 along the x and y axes, respectively. Additionally, the average number of persons per image is 1.16. For $D_{MS}^{train}$, the corresponding values are (330.2, 140.2) for the mean spatial location, with standard deviations of 166.3 and 11.95 along the x and y axes, respectively. The average number of persons per image is 1.03. In the following, we compare the loss values obtained via Equation (Equation 3) on both the training and test datasets for our proposed methods in Algorithms 1 and 2 across different settings. Since the used data were collected in the past, we compare the proposed method with the extrinsic parameters provided by FieldSAFE and ${\mathrm{MS}}^{2}$ using target-based calibration methods. For simplicity, we refer to them as ${\mathrm{FS}}_{\left[R,t\right]}$ and ${\mathrm{MS}}_{\left[R,t\right]}$.

In all our experiments, we used the hyper-parameters in Table 1 by default, unless another configuration was specified. We determined the hyper-parameters for the proposed algorithms through a process of testing various candidates and relying on intuition.

To compare Algorithms 1 and 2 with each other as well as with ${\mathrm{FS}}_{\left[R,t\right]}$ and ${\mathrm{MS}}_{\left[R,t\right]}$, in Table 2, we reported the Equation (Equation 3) loss values obtained by their corresponding R and t on the test datasets $D_{FS}^{test}$ and $D_{MS}^{test}$. As can be seen in the table, Algorithm 2, which uses outlier handling, obtains better results than Algorithm 1. Additionally, Algorithm 2 outperforms ${\mathrm{FS}}_{\left[R,t\right]}$ and ${\mathrm{MS}}_{\left[R,t\right]}$, which are obtained using calibration methods based on the target.

Figure 3 presents some performance metrics for Algorithm 2 optimized on $D_{FS}^{train}$. Figure 3 includes four plots, each displaying different aspects of the optimization process in each generation. All the loss values for the figure are computed using Equation (Equation 3). We just reported the plots for $D_{FS}^{train}$ as the representative of both the $D_{FS}^{train}$ and $D_{MS}^{train}$ datasets. Figure 3a shows the training loss value of the individual with the lowest training loss. Because of elitism, mutation, and crossover, the training loss value for the individual with the lowest training loss always remains non-increasing across generations. Figure 3b,c illustrate the log-average training loss of all individuals and the standard deviation of the loss among all individuals in the population. As individuals with lower training loss have a higher probability of being selected for crossover and mutation, increasing the number of generations results in a decrease in the log-average and standard deviation of train loss. However, due to randomness in mutation and crossover, these values eventually converge to a certain point and fluctuate around it. Finally, Figure 3d demonstrates the test loss of the individual with the lowest training loss. As depicted in the figure, both the training and test losses exhibit an initial exponential decrease, followed by a gradual convergence to a small value.

To assess the influence of the training dataset size on Algorithms 1 and 2, we performed the sub-sampling of $D_{FS}^{train}$ and $D_{MS}^{train}$, resulting in new training datasets ranging in size from 5 to the full dataset size, with a step size of 5. Since Algorithm 2 requires a minimum of 15 samples to determine a set of extrinsic parameters and subsequently test other samples for inlier status, we opted not to execute Algorithm 2 for configurations with 15 samples or fewer. As shown in Table 3 and its equivalent bar charts in Figure 4, increasing the number of data pairs for the training set from a small number decreases the test loss values significantly. Also, Algorithm 1 exhibits fluctuation in test loss values as the number of thermal images and point cloud pairs in the training set increases. In contrast, Algorithm 2 experiences fewer fluctuations. Additionally, in almost all cases, Algorithm 2 demonstrates superior performance compared with Algorithm 1 with the same training dataset size. In the case of 30 pairs in the dataset $D_{FS}^{train}$ and 20 pairs in the dataset $D_{MS}^{train}$, Algorithm 2 obtained a slightly worse result, which could be attributed to randomness, especially in the selection of a subsampled set from the dataset to assess the inlier or outlier status of non-subsampled data pairs. As the results in Table 3 suggest, not having a sufficient number of samples prevents us from executing the algorithms or obtaining good results.

To assess the robustness of Algorithms 1 and 2 under more extreme conditions, we generated $D_{FS-SW4}^{train}$ by swapping the thermal mask ($I_{t_{i}}^{h}$) for two random samples with another two random samples in $D_{FS}^{train}$. It caused four pairs of thermal images and LiDAR point clouds to lack matching masks in both modalities. Similarly, we created $D_{MS-SW4}^{train}$ using the same method. Furthermore, to investigate under different levels of mismatch, we generated comparable datasets by interchanging 4, 6, and 8 pairs, resulting in 8, 12, and 16 mismatched samples, respectively. As shown in Table 4 and its equivalent bar charts in Figure 5, Algorithm 2 achieved significantly better test loss and demonstrated greater robustness. In this experiment, $threshold_{sample}$ and $iter_{outlier}$ were set to three and five for new datasets derived from $D_{FS}^{train}$, and the variable $ratio_{solution}$ was set to 0.3 for $D_{MS-SW12}^{train}$ and $D_{MS-SW16}^{train}$. By increasing the number of mismatched pairs, the performance of both algorithms dropped; however, this effect was more significant for Algorithm 1. As the results suggest, it is critical to have good object detection in both modalities; otherwise, large amounts of false positives and false negatives from object detectors can degrade the quality of extrinsic parameters. Another interpretation could be that the presence of many people in a thermal image-point cloud pair may result in more mistakes in segmenting humans in both modalities due to a higher chance of overlapping. Therefore, selecting a large value for $H_{\max}$ may consequently lead to poorer results.

As mentioned earlier, it is important to collect a dataset with thermal images depicting humans in different locations and sizes. In order to assess the robustness of Algorithms 1 and 2 when dealing with highly unbalanced human locations in a collected dataset, we generated $D_{FS-NL}^{train}$ from $D_{FS}^{train}$ by removing samples where the human masks are located in the left one-third section of the image. $D_{FS-NL}^{train}$ comprises 27 samples. Similarly, we created $D_{MS-NR}^{train}$ by removing samples where the human masks are located in the right one-third of the image. $D_{MS-NR}^{train}$ consists of 36 samples. We generated these two imbalanced pose datasets from various imbalanced datasets that can be created to serve as a representative sample of this issue. As Table 5 shows, the mentioned unbalanced condition decreases performance when compared with the performance on a balanced dataset of a similar size in Table 3. However, Algorithm 2 is less affected by this in comparison with Algorithm 1. Therefore, it is important to have humans in diverse locations in the thermal camera’s field of view; otherwise, the pose imbalance can negatively affect the extrinsic calibration.

To assess the importance of each component in Algorithms 1 and 2, we systematically removed one component at a time and reported the results by calculating the Equation (Equation 3) loss using the test dataset $D_{FS}^{test}$, as shown in Table 6. The table reveals that removing elitism results in divergence and has the most significant impact. Subsequently, both the crossover and mutation exhibit notable importance, albeit to varying degrees. Removing the condition that projects 50% of the point cloud into the thermal image during the creation of the initial population has the least impact on test loss.

To observe the impact of the changes in certain hyper-parameters of Algorithms 1 and 2 and explain our intuition for selecting default values of hyper-parameters, we modified one parameter at a time while keeping all other parameters constant, as specified in Table 1. The corresponding results are presented in Table 7. In most cases, selecting values near the default showed no significant degradation in the performance of both algorithms. To demonstrate a more pronounced effect, we opted for more extreme values in comparison with the defaults. However, even in this scenario, in many cases, the results were not substantially different from the results of the default hyper-parameters.

As depicted in Table 7, a small population size ($N_{pop}$) results in poorer outcomes than the default value due to insufficient diversity. Conversely, a large population size slows down convergence and adds unnecessary computational overhead, approaching results similar to the default value. A low value of $pct_{elite}$ implies that many of the found good solutions do not directly transition to the next generation, diminishing their contribution to the overall population quality. Conversely, a large value of $pct_{elite}$ restricts the introduction of new individuals. In both cases, the results are inferior compared with the default value. A smaller value of $pct_{crossover}$ implies that fewer individuals in the next generation are produced by crossover, and more individuals are created by mutation. In the proposed algorithms, crossover covers a large area in the optimization space, and, as shown, a small value of $pct_{crossover}$ resulted in significantly poorer performance compared with the default value. In these algorithms, the mutation operation allows for the discovery of better solutions in the proximity of an existing solution. On the contrary, a large value of $pct_{crossover}$ means less mutation, leading to lower performance compared with the default values. Finding a balance between the crossover and mutation is crucial for achieving good results. $\sigma_{rot}$ and $\sigma_{trans}$ represent the noise levels for the mutation operator, determining how much change in a found solution is applied to generate a new individual. A very small amount does not alter parameters in the optimization space enough to produce a meaningful change in the outcome, while a large amount results in an individual that is very different from the original solution and does not retain its attributes. As shown, in both cases, the results are worse than the default values.

A low value of $threshold_{sample}$ imposes a stringent criterion for considering a sample in the dataset as an inlier, potentially causing issues by incorrectly classifying many good pairs in the data as outliers and rejecting them from the calculation of extrinsic parameters. Conversely, a high-value results in the ineffective detection of outlier samples in data. In both cases, the performance is weaker compared with the default value. As depicted in Table 3, augmenting the pairs for optimizing extrinsic parameters generally leads to improved performance. A small value of $threshold_{sample}$ results in the identification of suboptimal extrinsic parameters, leading to poor outlier detection performance. Conversely, when the value of $threshold_{sample}$ is large, there is a higher likelihood of including a significant amount of outliers. The algorithm may face challenges in identifying a robust model amidst the abundance of irrelevant data. As indicated in Table 7, in both scenarios, the performance is diminished compared with the default value. We selected the default value for $min_{sample}$, as represented in Table 1, based on the performance of Algorithm 1 in Table 3. As shown in Table 7, a low value for $min_{sample}$ can result in obtaining a poor initial estimate for extrinsic calibration parameters, thereby impacting the performance of determining inliers. Additionally, a large value can lead to the exclusion of a significant number of samples from the determination of whether they are outliers or not, resulting in poorer results. As can be interpreted from Table 7, a small value for $iter_{outlier}$ can cause many samples not to be examined for being outliers, resulting in a decrease in performance. On the other hand, a large value does not contribute to finding more outliers, and the performance remains similar to a balanced $iter_{outlier}$ while only increasing computation. As indicated by the values in Table 7, a low value of $ratio_{solution}$ does not alter the performance in the specific experiment of $D_{FS}^{test}$. However, a high value of $ratio_{solution}$ led to poor performance, as the proportion of inliers in each iteration of Algorithm 2 was smaller than the $ratio_{solution}$, and, consequently, the detected outliers were rejected.

In Figure 6, the dots represent projected points in the LiDAR point cloud onto a thermal image using a set of R and t. This figure presents a qualitative comparison of Algorithm 2 (blue dots) with ${\mathrm{FS}}_{\left[R,t\right]}$ and ${\mathrm{MS}}_{\left[R,t\right]}$ (red dots) on two frames from $D_{FS}^{test}$ and $D_{MS}^{test}$. As can be observed, both the red and blue dots are closely aligned, demonstrating that our proposed algorithm and ${\mathrm{FS}}_{\left[R,t\right]}$ and ${\mathrm{MS}}_{\left[R,t\right]}$ are in close agreement. However, as depicted in the zoomed-in patches in Figure 6b,d, the blue projected points that correspond to humans in the point cloud are more closely aligned with the humans in the thermal images. Additionally, in Figure 6d, the blue points are more centered on the streetlight.

## 5. Conclusions and Future Work

In this paper, we have highlighted the advantages of combining data from thermal cameras and LiDAR sensors and emphasized the importance of accurately determining the rotation matrix R and the translation vector t to effectively utilize data from both the thermal camera and LiDAR. Also, we mentioned certain challenges associated with using specific targets visible in thermal cameras, especially when dealing with regular sensor drift or changing settings. To address these challenges, we have introduced an extrinsic calibration algorithm. This algorithm aligns a thermal camera and a LiDAR without the need for a dedicated target. This calibration is achieved by matching segmented human subjects in both modalities using pairs of thermal images and LiDAR point clouds that were collected during the sensor setup’s movement. Firstly, we introduced the procedure for constructing a dataset comprising pairs, where each pair consists of thermal camera data and its corresponding point cloud. Secondly, we presented a novel loss function that quantifies the alignment between the LiDAR and thermal camera coordinate systems given the rotation matrix R and translation vector t. Thirdly, we introduced two evolutionary algorithms, one of which does not explicitly address the issue of outliers, while the other mitigates the impact of outliers. Also, our proposed algorithm obviates the need for an initial estimate of R and t. Finally, we conducted a series of comprehensive experiments to assess the efficiency of the proposed algorithms under various settings and to compare the performance of them with the provided extrinsic parameters in the FieldSAFE dataset [11] and the ${\mathrm{MS}}^{2}$ dataset [13]. This comparison offers a quantitative and qualitative assessment of our method’s performance, providing valuable insights into its effectiveness and robustness. In one instance, our method exhibits a noteworthy 4.43% improvement in the designed loss compared with extrinsic parameters derived from target-based calibration in the FieldSAFE dataset. In another instance, distorting a dataset by randomly swapping thermal cameras of four pairs in the data with another four pairs to create a new dataset with eight mismatches between thermal images and point clouds only resulted in an 8.7% increase in the loss, showcasing its robustness.

For future work, we plan to explore several directions based on the different experiments presented. Firstly, we aim to achieve better results with fewer pairs in the dataset. Secondly, as demonstrated, the dataset collected from thermal cameras indicates that humans are often not in varying positions, and distances from the camera can negatively impact the quality of the extrinsic calibration. We will investigate methods, such as weighting different pairs, to address this issue. Thirdly, we will explore multi-objective optimization to incorporate more complex information about masked humans in both modalities in order to obtain better results.

## Figures and Tables

**Figure 1 sensors-24-00669-f001:**
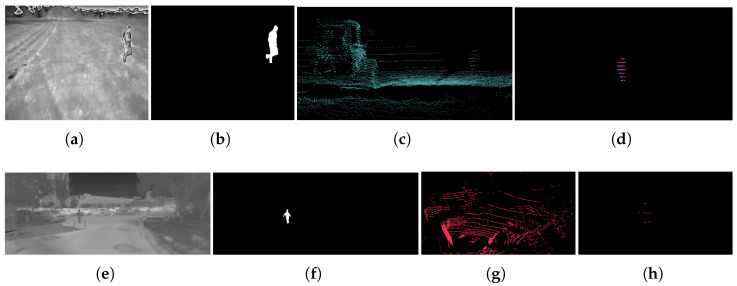
Images (**a**–**d**) are sourced from the FieldSAFE dataset [11], whereas images (**e**–**h**) are obtained from the ${\mathrm{MS}}^{2}$ dataset [13]. In each row, the images from left to right show a thermal image ($I_{t_{i}}$), the segmentation mask for human(s) in the thermal image ($I_{t_{i}}^{h}$), a shot from its corresponding point cloud ($P_{t_{i}}$), and a shot from the corresponding point cloud with only human(s) points ($P_{t_{i}}^{h}$).

**Figure 2 sensors-24-00669-f002:**
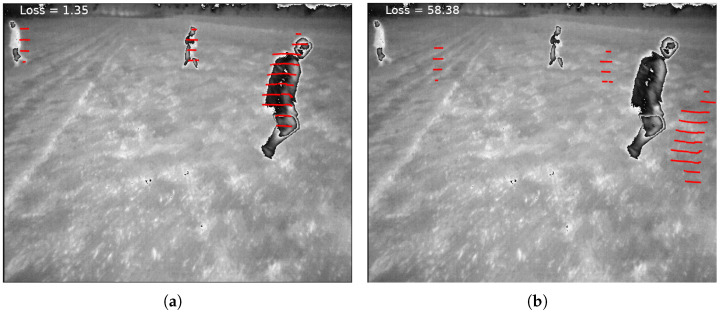
Images (**a**) and (**b**) show the projection of a point cloud onto a thermal image for a sample pair from the FieldSAFE dataset [11] with two different sets of R and t. Equation (Equation 1) loss value for the extrinsic parameters used in image (**a**) is 1.35, while the loss value for the extrinsic parameters used in image (**b**) is 58.38.

**Figure 3 sensors-24-00669-f003:**
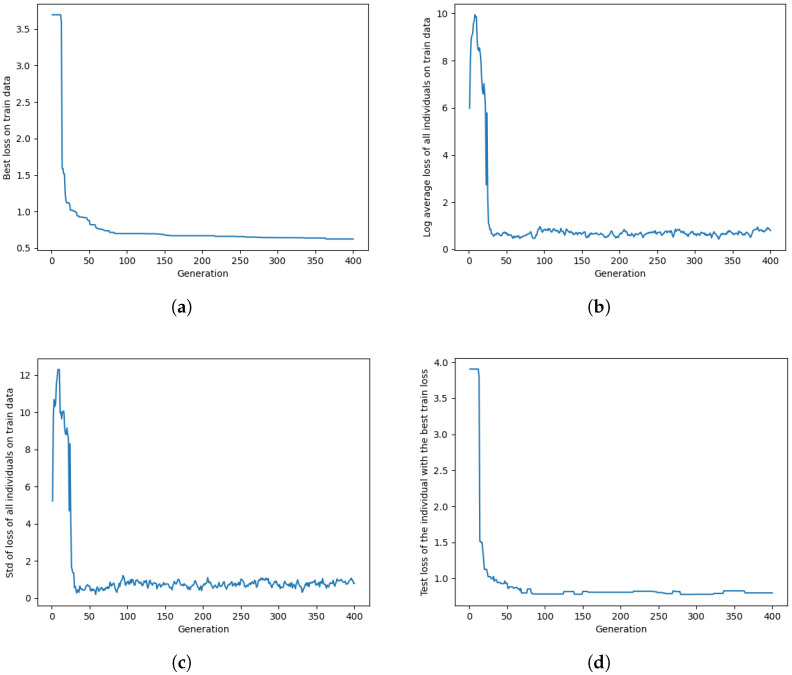
Plots for Algorithm 2 optimized on $D_{FS}^{train}$ depicting (**a**) the train loss of the individual with the lowest train loss in each generation, (**b**) the log-average train loss of all individuals in the population in each generation, (**c**) the standard deviation of the loss among all individuals in the population for each generation, and (**d**) the test loss of the individual with the lowest train loss in each generation.

**Figure 4 sensors-24-00669-f004:**
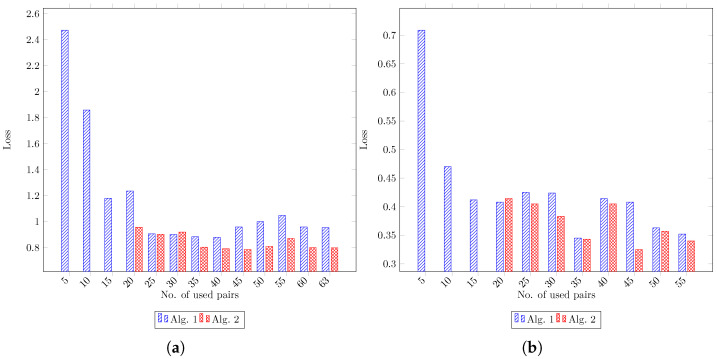
(**a**,**b**) are bar charts for datasets derived by subsampling from $D_{FS}^{train}$ and $D_{MS}^{train}$, respectively, as created from Table 3. They display the test loss values of Algorithms 1 and 2 calculated by Equation (Equation 3) on $D_{FS}^{test}$ and $D_{MS}^{test}$.

**Figure 5 sensors-24-00669-f005:**
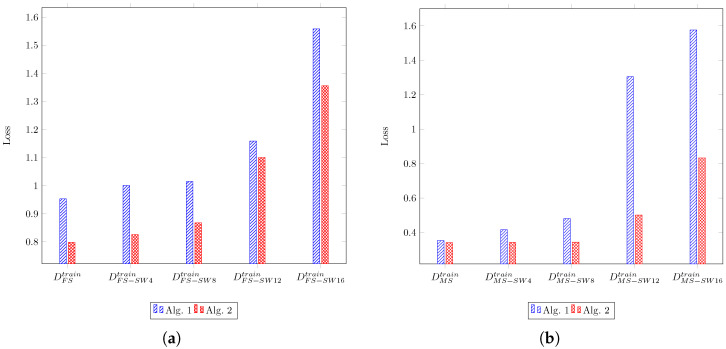
(**a**,**b**) are bar charts, respectively, for datasets derived from $D_{FS}^{train}$ and $D_{MS}^{train}$ by swapping thermal masks. Bar charts (**a**,**b**) are created from Table 4. The provided values correspond to the losses computed using Equation (Equation 3) on $D_{FS}^{test}$ and $D_{MS}^{test}$.

**Figure 6 sensors-24-00669-f006:**
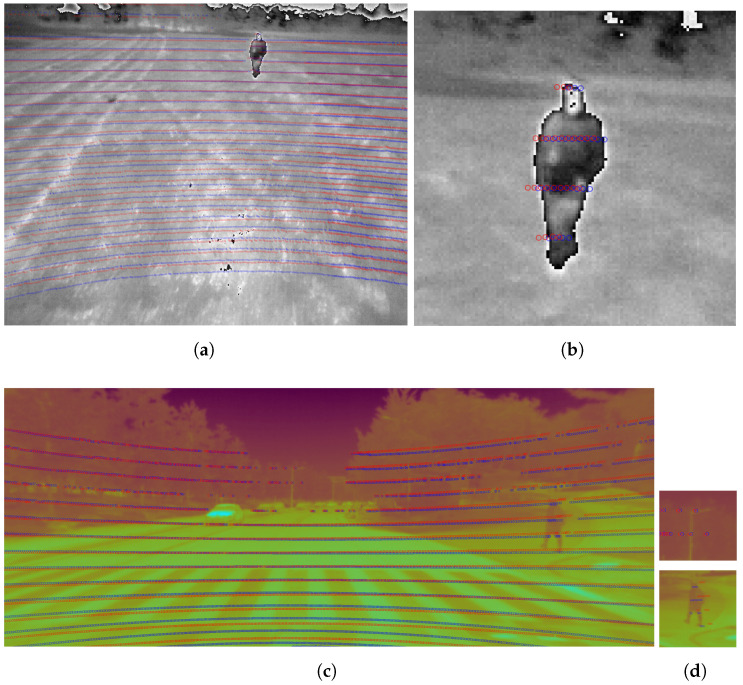
Images (**a**,**c**) respectively show a comparison of Algorithm 2 (blue dots) with ${\mathrm{FS}}_{\left[R,t\right]}$ and ${\mathrm{MS}}_{\left[R,t\right]}$ (red dots) on two samples from FieldSAFE [11] and ${\mathrm{MS}}^{2}$ [13] datasets. The dots represent projected points from the LiDAR point cloud onto the thermal image. Additionally, the images (**b**,**d**) are zoomed-in patches taken from the frames on (**a**) and (**c**), respectively. To enhance visual interpretation, the image in (**c**) and its zoomed-in patches in (**d**) were pseudocolored from the original grayscale image.

**Table 1 sensors-24-00669-t001:** Hyper-parameters for Algorithms 1 and 2.

Hyper-Parameter	Value
$N_{pop}$	500
$iter_{max}$	400
$interval_{rot}$	[−3.5, 3.5] rad
$interval_{trans}$	[−1000, 1000] mm
$pct_{elite}$	15%
$pct_{crossover}$	40%
$c_{1}$	5
$c_{2}$	5
$min_{sample}$	20ifnumberoftrainsample≥40;else,15
$iter_{outlier}$	2
$ratio_{solution}$	0.7
$threshold_{sample}$	FieldSAFE: 2.0, ${\mathrm{MS}}^{2}$: 1.5
$\sigma_{rot}$	0.02 rad
$\sigma_{trans}$	20 mm

**Table 2 sensors-24-00669-t002:** Comparison of Equation (Equation 3) loss for different methods on $D_{FS}^{test}$ and $D_{MS}^{test}$ datasets.

		Dataset	
		$D_{\mathrm{FS}}^{\mathrm{test}}$	$D_{\mathrm{MS}}^{\mathrm{test}}$
Method	Algorithm 1	0.953	0.352
	Algorithm 2	**0.798**	**0.34**
	${\mathrm{FS}}_{\left[R,t\right]}$	0.835	-
	${\mathrm{MS}}_{\left[R,t\right]}$	-	2.731

**Table 3 sensors-24-00669-t003:** The effect of varying the size of the training dataset on the test loss values of Algorithms 1 and 2. The reported loss values calculated by Equation (Equation 3) on $D_{FS}^{test}$ and $D_{MS}^{test}$.

	No. of Used Pairs	5	10	15	20	25	30	35	40	45	50	55	60	63
$D_{FS}^{test}$	Algorithm 1	2.474	1.858	1.179	1.236	0.905	0.9	0.884	0.878	0.959	0.999	1.047	0.959	0.953
	Algorithm 2	-	-	-	0.957	0.902	0.919	0.804	0.791	0.785	0.808	0.869	0.799	0.798
$D_{MS}^{test}$	Algorithm 1	0.709	0.47	0.412	0.408	0.425	0.424	0.345	0.414	0.408	0.363	0.352	-	-
	Algorithm 2	-	-	-	0.414	0.405	0.383	0.343	0.405	0.325	0.357	0.34	-	-

**Table 4 sensors-24-00669-t004:** Comparing Algorithms 1 and 2’s test loss under harsher conditions by introducing artificial mismatches between masks in both modalities. The provided values correspond to the loss values computed using Equation (Equation 3) on $D_{FS}^{test}$ and $D_{MS}^{test}$.

	Algorithm 1	Algorithm 2
$D_{FS}^{train}$	0.953	0.798
$D_{FS-SW4}^{train}$	1.001	0.826
$D_{FS-SW8}^{train}$	1.015	0.868
$D_{FS-SW12}^{train}$	1.159	1.100
$D_{FS-SW16}^{train}$	1.558	1.356
$D_{MS}^{train}$	0.352	0.340
$D_{MS-SW4}^{train}$	0.415	0.342
$D_{MS-SW8}^{train}$	0.480	0.343
$D_{MS-SW12}^{train}$	1.305	0.500
$D_{MS-SW16}^{train}$	1.577	0.832

**Table 5 sensors-24-00669-t005:** Comparing Algorithms 1 and 2’s test loss values calculated using Equation (Equation 3) on $D_{FS}^{test}$ and $D_{MS}^{test}$ under unbalanced human locations in a collected dataset.

	Algorithm 1	Algorithm 2
$D_{FS-NL}^{train}$	1.482	1.415
$D_{MS-NR}^{train}$	0.463	0.356

**Table 6 sensors-24-00669-t006:** The effect of removing different components from Algorithms 1 and 2 on the loss of Equation (Equation 3) on the dataset $D_{FS}^{test}$.

Removed Part	None	Elitism	Mutation	Crossover	No Init. Const.
Algorithm 1	0.953	3756.001	1.596	8.082	0.972
Algorithm 2	0.798	3391.615	1.595	23.626	0.928

**Table 7 sensors-24-00669-t007:** The effect of changing some of the default hyper-parameters on Algorithms 1 and 2 on the loss of Equation (Equation 3) on the dataset $D_{FS}^{test}$.

Hyper-Parameter	All	$N_{pop}$		$pct_{elite}$		$pct_{crossover}$		$\sigma_{rot}$		$\sigma_{trans}$	
Value	Default	100	800	2%	60%	10%	90%	0.005	0.2	5	200
Algorithm 1	0.953	1.588	0.962	2.209	0.97	1.011	1.043	1.021	1.023	1.509	1.049
Hyper-Parameter	All	${threshold}_{sample}$		${min}_{sample}$		${iter}_{outlier}$		${ratio}_{solution}$		-	
Value	Default	0.5	6.0	10	30	1	5	0.1	0.9	-	
Algorithm 2	0.798	0.952	0.845	0.957	0.84	0.95	0.8	0.798	0.95	-	

## Data Availability

Data are contained within the article.

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
