# Peer review of "Extrinsic Calibration of Thermal Camera and 3D LiDAR Sensor via Human Matching in Both Modalities during Sensor Setup Movement"

_sensors, 2024, doi:10.3390/s24020669_

Round 1

Reviewer 1 Report

Comments and Suggestions for Authors

In this manuscript, the authors describe two sensor fusion algorithms for LIDAR and thermal camera. Their methods use human presence to infer the relative translation and rotation of the two sensors without dedicated calibration targets. The authors test their methods using open datasets and conclude that the methods are reliable and efficient. This work is interesting and the manuscript is well-organized. I recommend its publication if the following issues are addressed.

1    1)  The authors mentioned the sizes and the locations of humans in the scene affect the calibration results. Please specify the average number and spatial distribution of humans in the used datasets. How many humans are required to have a reliable calibration generally?

2    2)  Based on 1), please comment on how the methods perform in scenarios with many versus few human subjects.

3    3)  If the LIDAR points and the thermal camera data also have scale difference, would these methods still be robust?

Author Response

Dear Editor and reviewers,

Your dedication and the time you spent reviewing the article are truly appreciated. Thank you for sharing valuable insights; we are grateful for your contribution.

All modifications in the newly uploaded article are indicated in red or highlighted in yellow.

Editor:

During the modification stage, please re-edit Section 2 to reduce duplication.

In Section 2, several parts were modified to minimize duplication.

Reviewer 1:

- The authors mentioned the sizes and the locations of humans in the scene affect the calibration results. Please specify the average number and spatial distribution of humans in the used datasets. How many humans are required to have a reliable calibration generally?

These details were added (starting from line 425). At least one human per thermal image-point cloud pair is required, and reliable results can be obtained with only one person. Also, as indicated in Table 3, around 30 pairs of thermal image-point cloud pairs can yield reliable results.

- Based on 1), please comment on how the methods perform in scenarios with many versus few human subjects.

These details were added (starting from lines 225, 424, 489). We only retained thermal image-LiDAR point cloud pairs with at most three persons in them, as a larger number of people increases the chance of partial overlapping and potentially can cause more errors in segmenting humans in both modalities. Additionally, the datasets used often have at most three people in each image. However, from the results in Table 4, it can be interpreted that having more people can have a negative effect on performance.

- If the LIDAR points and the thermal camera data also have scale difference, would these methods still be robust?

The details were added (starting from line 370). The added paragraph is as follows: The proposed algorithms aim to determine a rigid body transform between the coordinate systems of a thermal camera and a LiDAR sensor by estimating the rotation matrix R and translation vector t. It is essential for both sensors to operate with the same scale for accurate results. If the two sensors are not on the same scale, and assuming the factory configurations of sensors are available (which is almost always the case for these two types of sensors), this information can be used to preprocess the data and convert them to the same scale. In Eq. 1, K(RpL + t) is utilized to map a LiDAR point in the image coordinate system in a homogeneous format. Subsequently, the homogeneous point is converted to an inhomogeneous coordinate in the thermal image. When using data with different scales, as the cost function minimizes the distance in the thermal image, it can yield a solution that effectively maps LiDAR points to their corresponding thermal image pixels, even when dealing with data of varying scales. However, the obtained translation vector may not accurately represent the real distance between the sensors, as it will be scaled by the difference in scale between the two sensors.

Reviewer 2 Report

Comments and Suggestions for Authors

The overall structure of the paper is clear, and the experimental results are abundant, but there are still several problems that need to be elaborated and modified:

1. The authors should compare their proposed algorithm with other heuristic algorithms to evaluate its effectiveness in the extrinsic calibration of thermal camera and 3D LiDAR sensor via human matching in both modalities during sensor setup movement.

2.  The parameters  in the algorithm been thoroughly discussed and experimentally validated, such as parameters :

minsample  , iteroutlier ,ratiosolution and thresholdsample.

Comments on the Quality of English Language

The overall English writing of the paper is relatively standardized, and only a few grammatical errors need to be corrected.

Author Response

Dear Editor and reviewers,

Your dedication and the time you spent reviewing the article are truly appreciated. Thank you for sharing valuable insights; we are grateful for your contribution.

All modifications in the newly uploaded article are indicated in red or highlighted in yellow.

Editor:

During the modification stage, please re-edit Section 2 to reduce duplication.

In Section 2, several parts were modified to minimize duplication.

Reviewer  2:

- The authors should compare their proposed algorithm with other heuristic algorithms to evaluate its effectiveness in the extrinsic calibration of thermal camera and 3D LiDAR sensor via human matching in both modalities during sensor setup movement.

The details were added (starting from line 370). In the article, we compare the proposed algorithms with two calibration-based methods. To the best of our knowledge, this work is the first to utilize humans during sensor setup movement to calibrate both the thermal camera and LiDAR. In Section 2, the related work, we initially mentioned two targetless methods for extrinsic calibration of the thermal camera and LiDAR. We have updated their descriptions in the related work for greater clarity.

Fu et al. [28] introduced a targetless extrinsic calibration method that calibrates a stereo visual camera system, a thermal camera, and a LiDAR sensor. The thermal camera is calibrated with the left camera in the stereo system by simultaneously using data from LiDAR and the left stereo camera. By establishing transformations between the thermal camera and the stereo system, as well as between LiDAR and the stereo system, the transformation between LiDAR and the thermal camera can be calculated. However, their method requires a good initial hyperparameter. In contrast, our method utilizes only the thermal camera and LiDAR sensor, unlike their approach that simultaneously employs stereo vision cameras, a thermal camera, and LiDAR.

In Mharolkar et al. [30], targetless cross-calibration was proposed for visual and thermal cameras with LiDAR sensors by utilizing a deep neural network. Instead of using sensor data directly or handcrafted features, deep features were employed. To train the network for calibrating the visual camera and LiDAR on the KITTI360 dataset [31], they utilized 44,595 image-point cloud pairs. For training the network for calibrating the thermal camera, pre-trained weights for the visual camera and LiDAR were employed, and the model was trained on their thermal camera and LiDAR dataset, consisting of 8,075 private thermal image and LiDAR pairs. Additionally, for a new set of sensors, the network should be re-trained. In their work, they used 44,595 pairs of RGB color-point clouds to pre-train their model and applied transfer learning on a private dataset of 8,057 thermal image-point cloud pairs. We only utilized thermal images and point clouds in our method, and there is no data from other modalities.

For these reasons, we compared the proposed method with FS[R,t] and MS[R,t]  methods.

- The parameters in the algorithm been thoroughly discussed and experimentally validated, such as parameters : minsample , iteroutlier ,ratiosolution and thresholdsample.

The details were added (starting from line 553). We added 8 new experiments to Table 7 related to these four hyper-parameters, and we also included the interpretation of the results.

- The overall English writing of the paper is relatively standardized, and only a few grammatical errors need to be corrected.

We re-read the article, found several grammar errors and fixed them. Also, there were several double spaces, which we fixed.
